# Association between Dysfunction of the Nucleolar Stress Response and Multidrug Resistance in Pediatric Acute Lymphoblastic Leukemia

**DOI:** 10.3390/cancers14205127

**Published:** 2022-10-19

**Authors:** Shunsuke Nakagawa, Kohichi Kawahara, Yasuhiro Okamoto, Yuichi Kodama, Takuro Nishikawa, Yoshifumi Kawano, Tatsuhiko Furukawa

**Affiliations:** 1Department of Pediatrics, Graduate School of Medical and Dental Sciences, Kagoshima University, Kagoshima 890-8520, Japan; 2Department of Molecular Oncology, Graduate School of Medical and Dental Sciences, Kagoshima University, Kagoshima 890-8520, Japan

**Keywords:** B-cell precursor acute lymphoblastic leukemia, pediatric leukemia, P53, nucleolar stress response, multidrug resistance, relapse, refractoriness, chemotherapy

## Abstract

**Simple Summary:**

Pediatric patients with B-cell precursor acute lymphoblastic leukemia (BCP-ALL) relapse or are refractory to chemotherapy despite the low frequency of *TP53* mutations. The nucleolar stress response is one of the mechanisms that activate P53 by ribosomal protein L11 (RPL11). We hypothesized that the lack of nucleolar stress response is related to chemoresistance and relapse in some pediatric BCP-ALL cases. We revealed that clinical BCP-ALL therapeutics, such as 6-mercaptopurine, methotrexate, daunorubicin, and cytarabine, induced the nucleolar stress response, and its treatment susceptibility was dependent on the nucleolar stress response. Furthermore, we observed decreased RPL11 expression at relapse in seven children with BCP-ALL in comparison to that at onset. Our findings provide new insights into the anti-leukemia mechanism in BCP-ALL and multidrug resistance and relapse via the nucleolar stress response, suggesting that the nucleolar stress response may be a potential therapeutic strategy to predict chemosensitivity and improve chemoresistance in pediatric BCP-ALL.

**Abstract:**

Approximately 20% of pediatric patients with B-cell precursor acute lymphoblastic leukemia (BCP-ALL) relapse or are refractory to chemotherapy despite the low frequency of *TP53* mutations. The nucleolar stress response is a P53-activating mechanism via MDM2 inhibition by ribosomal protein L11 (RPL11). We analyzed the role of the nucleolar stress response using BCP-ALL cell lines and patient samples by drug sensitivity tests, Western blotting, and reverse transcription polymerase chain reaction. We revealed that the nucleolar stress response works properly in *TP53* wild-type human BCP-ALL cell lines. Next, we found that 6-mercaptopurine, methotrexate, daunorubicin, and cytarabine had anti-leukemic effects via the nucleolar stress response within BCP-ALL treatment. Comparing the samples at onset and relapse in children with BCP-ALL, *RPL11* mRNA expression decreased at relapse in seven of nine cases. Furthermore, leukemia cells with relapse acquired resistance to these four drugs and suppressed P53 and RPL11 expression. Our findings suggest that the nucleolar stress response is a novel anti-leukemia mechanism in BCP-ALL. As these four drugs are key therapeutics for BCP-ALL treatment, dysfunction of the nucleolar stress response may be related to clinical relapse or refractoriness. Nucleolar stress response may be a target to predict and improve the chemotherapy effect for pediatric BCP-ALL.

## 1. Introduction

The tumor suppressor *TP53* encodes the P53 protein, which inhibits cell growth and induces cell cycle arrest and apoptosis [1]. Nucleolar/ribosomal stress, including inhibition of rRNA synthesis, processing, or ribosome subunit assembly, increases P53 signaling via ribosomal proteins (RPs), including RPL5, RPL11, and RPL23. In this nucleolar stress response, the nucleolus is disassembled, and RPs are consequently released from the nucleolus to the nucleoplasm, where they bind to and suppress MDM2. As a result, the response stabilizes and increases P53 [2,3,4]. To contribute to this anti-tumor mechanism, *TP53* must be of wild-type. Actinomycin D or 5-fluorouracil are well known to activate the nucleolar stress response in solid adult tumors [5]. We have reported that the low activity of the nucleolar stress response is related to poorer prognosis in colorectal, esophageal, gastric, and lung cancers [6,7,8,9].

The survival of patients with pediatric B-cell precursor acute lymphoblastic leukemia (BCP-ALL) has been improved remarkably, with 5-year survival rates of 85–90% in developed countries [10]. Risk-adopted multidrug combination chemotherapy and hematopoietic stem cell transplantation have contributed to successful outcomes. However, almost 20% of patients relapse or have cells resistant to chemotherapy. In pediatric BCP-ALL cases, some gene mutations have been found to be a drug-resistant factor at relapse; however, multidrug resistance has not been reported. Therefore, it is necessary to improve the survival of pediatric BCP-ALL by developing new anti-tumor agents or novel treatment strategies that can overcome multidrug resistance. To the best of our knowledge, there are no previous reports on the function of the nucleolar stress response in leukemia.

Mutations in *TP53* were found in 50% of human cancers [11] and were related to poor prognosis [12]. In pediatric BCP-ALL, *TP53* mutations were reportedly associated with poor prognosis [13]. Evidence indicates that P53 has an anti-leukemia effect in pediatric BCP-ALL cases. However, the relevance of *TP53* somatic mutations in pediatric BCP-ALL is only 2–4% at diagnosis and 12% at relapse. Owing to this low rate of *TP53* mutation in refractory or relapsed BCP-ALL, P53 is expected to be inactivated as it acquires abnormalities in the mechanisms regulating it (as opposed to through *TP53* mutations).

We hypothesized that, in pediatric BCP-ALL, the nucleolar stress response works as an anti-leukemia mechanism and that its dysfunction is related to relapse or resistance to chemotherapy.

## 2. Materials and Methods

### 2.1. Patients and Samples

Bone marrow samples containing leukemia cells were obtained from pediatric patients with BCP-ALL. All patients were treated at Kagoshima University Hospital between 2005 and 2020. Mononuclear cells were isolated using sucrose density-gradient centrifugation (density, 1.077 g/mL; Lymphoprep^TM^, Alere Technologies, Waltham, MA, USA) and stored at −130 °C. For each assay, leukemia cells were resuspended in RPMI-1640 (Fujifilm Wako Pure Chemical Corporation, Osaka, Japan) supplemented with 20% fetal bovine serum (FBS), 5 μg of insulin per mL, 5 μg of transferrin per mL, and 5 ng of sodium selenite per mL (ITS media supplement; Sigma-Aldrich, St. Louis, MO, USA) [14]. Leukemia cells were not isolated and purified from bone marrow samples before being used for each assay. This study was approved by the ethics committee of the Kagoshima University Graduate School of Medical and Dental Sciences (approval number: 170242) and was conducted in accordance with the principles of the Declaration of Helsinki.

### 2.2. Cell Culture and Reagents

NALM6 cells and the RS4;11 *TP53* wild-type BCP-ALL cell line were obtained from the JCRB Cell Bank (Osaka, Japan) and the American Type Culture Collection (Manassas, VA, USA), respectively. These cells were cultured in RPMI-1640 (Fujifilm Wako Pure Chemical Corporation, Osaka, Japan) containing 10% heat-inactivated FBS (Thermo Fisher, Waltham, MA, USA), in a humidified incubator at 37 °C under 5% CO_2_.

The following chemicals were purchased: actinomycin D, daunorubicin, dimethyl sulfoxide (DMSO), methotrexate, prednisolone, vincristine, cytarabine, L-asparaginase, 6-mercaptopurine, and HDM201 (Cayman Chemical, Ann Arbor, MI, USA; Fujifilm Wako Pure Chemical Corporation; MedChemExpress, Monmouth Junction, NJ, USA; ProSpec, Ness-Ziona, Israel; TCI, Tokyo, Japan).

### 2.3. Quantitative Real-Time Polymerase Chain Reaction (PCR)

Total cellular RNA was extracted using the RNeasy^®^ Mini Kit (QIAGEN, Valencia, CA, USA). First-strand cDNA was synthesized using the PrimeScript^TM^ RT Reagent Kit (Perfect Real Time; Takara Bio Inc., Kusatsu, Japan) according to the manufacturer’s protocol. The *RPL11*, *P21*, *MDM2*, *Fas*, and *45S pre-rRNA* mRNA levels were quantified using TB Green^®^Premix Ex Taq^TM^ II (Takara Bio Inc.) according to the manufacturer’s protocol. Primers were as follows: RPL11 (Forward); 5′-GAAAAGGAGAACCCCATGC-3′, RPL11 (Reverse); 5′-CATTTCTCCGGATGCCAA-3′, P21 (Forward); 5′-TGAGCCGCGACTGTGATG-3′, P21 (Reverse); 5′-GTCTCGGTGACAAAGTCGAAGTT-3′, MDM2 (Forward); 5′-AGCCTCCAATGAGAGCAACTTGA-3′, MDM2(Reverse); 5′-CAGGCTGCCATGTGACCTAAGA-3′, Fas (Forward); 5′-TCTGCCATAAGCCCTGT-3′, Fas (Reverse); 5′-GTCTGTGTACTCCTTCCCT-3′, 45S pre-rRNA (Forward); 5′-CTCCGTTATGGTAGCGCTGC-3′, 45S pre-rRNA (Reverse); 5′-GCGGAACCCTCGCTTCTC-3′, GAPDH (Forward); 5′-GCACCGTCAAGGCTGAGAAC-3′, GAPDH (Reverse); 5′-TGGTGAAGACGCCAGTGGA-3′. The gene expression levels were normalized with respect to GAPDH. All reactions were performed in triplicate.

### 2.4. Immunoblotting Analysis

Immunoblotting was performed using a standard protocol with primary antibodies against P53 (DO-1), RPL11 (D1P5N), and GAPDH (14C10) (Cell Signaling Technology, Danvers, MA, USA), as well as MDM2 (SMP14; Santa Cruz Biotechnology Inc., Dallas, TX, USA). The proteins were detected using horseradish peroxidase-conjugated secondary antibodies (Cell Signaling Technology). The GAPDH levels were used as the loading controls.

### 2.5. Transfection with siRNA

siRNA sequences targeting RPL11 in addition to the scramble sequence were as follows: siRPL11#1: 5′-GGUGCGGGAGUAUGAGUUA-3′, siRPL11#2: 5′-GGAGUAUGAGUUAAGAAAA-3′, and scramble siRNA: 5′-UUCUCCGAACGUGUCACGU-3′. siRPL11#1 and scramble were synthesized by FASMAC (Kanagawa, Japan), and siRPL11#2 was synthesized by Ambion (Austin, TX, USA). The siRNAs were transfected into cells by electroporation using an AMAXA 4D-Nucleofector (Lonza, Basel, Switzerland) according to the manufacturer’s instructions.

### 2.6. Drug Resistance and Apoptosis

Cells transfected with siRNA were seeded in 384-well plates at a density of 1750 cells per well. After 24 h, each drug stimulated the cells for 72 h in NALM6 cells, 96 h in RS4;11 cells, and 96 h in the patients’ leukemia cells. Cell viability was calculated based on the transfected cells treated with the same concentration of DMSO or water as the drug. Viability was assessed using Cellno ATP ASSAY reagent ver.2 (TOYO B-NET, Tokyo, Japan).

The IC_50_ value was extrapolated from the growth inhibition curve produced for each drug and calculated using GraphPad Prism software (version 9.1.1; GraphPad Software Inc., San Diego, CA, USA).

To estimate the tendency of NALM6 cells to undergo apoptosis, caspases 3 and 7 activities were determined using the Caspase Glo^®^ 3/7 assay (Promega Corporation, Madison, WI, USA).

ATP luciferase activity and caspase 3/7 activity were evaluated using a luminescence assay conducted via a TriStar LB 941 multimode microplate reader (Berthold Technologies GmbH & Co., Bad Wildbad, Germany).

### 2.7. Immunoprecipitation

To detect the binding of MDM2 to RPL11, NALM6 cells were lysed with M-PER Mammalian Protein Extraction Reagent (Thermo Fisher) after adding 6-mercaptopurine, methotrexate, cytarabine, daunorubicin, and actinomycin D, and incubating with antibodies against MDM2 (Merck, Darmstadt, Germany) or normal mouse immunoglobulin G (IgG) (Santa Cruz Biotechnology Inc.). Immune complexes were adsorbed onto protein G-conjugated magnetic beads (Bio-Rad Laboratories, Inc., Hercules, CA, USA). After washing extensively, we analyzed the samples by immunoblotting with antibodies against RPL11 and MDM2.

### 2.8. Immunofluorescence

NALM6 and RS4;11 cells treated with each drug were allowed to adhere to polylysine-L coverslips (MATSUNAMI, Osaka, Japan) and were fixed in 4% paraformaldehyde at room temperature. Then, the cells were permeabilized with 0.5% TritonX for 10 min, washed in phosphate-buffered saline, stained with anti-nucleophosmin antibody (B23; Santa Cruz Biotechnology Inc.), and revealed with the appropriate secondary antibody coupled with Alexa Fluor^TM^ 594 (Thermo Fisher). DNA was counterstained with DAPI (ProLong Gold; Thermo Fisher). Cells were examined independently using a fluorescence microscope (BZ-X710; Keyence, Osaka, Japan).

### 2.9. Statistical Analyses

Data are presented as means ± standard deviations. We performed statistical analyses using Student’s *t*-test for mRNA expression tests. Analyses were performed using GraphPad Prism software (version 9.1.1).

## 3. Results

### 3.1. The Nucleolar Stress Response as an Anti-Leukemia Mechanism

To determine whether the nucleolar stress response exhibits an anti-tumor effect, we investigated the function of *TP53* wild-type BCP-ALL cells using actinomycin D. HDM201, an MDM2 inhibitor, was used as a negative control to demonstrate anti-tumor effects by increasing P53 regardless of the nucleolar stress response. RPL11 expression was suppressed by siRNA for RPL11 (siRPL11#1 and siRPL11#2) in NALM6 and RS4;11. NALM6 and RS4;11 cells with RPL11 knockdown showed resistance to actinomycin D compared to cells transfected with scramble siRNA. However, HDM201 did not show any statistically significant difference between cells with RPL11 knockdown and cells transfected with scramble siRNA (Figure 1a). In addition, the untreated NALM6 and RS4;11 cells were almost completely viable at 72 and 96 h. Caspase 3/7 activity showed that RPL11 knockdown in NALM6 and RS4;11 cells suppressed the apoptosis induced by actinomycin D; however, it did not suppress the apoptosis induced by HDM201 (Figure 1b). Actinomycin D increased the levels of P53 protein and its downstream genes, *P21*, *MDM2*, and *Fas*, in NALM6 cells and RS4;11 cells transfected with scramble siRNA, but not in the cells with RPL11 knockdown ( Figure 1c,d and Appendix A). Thus, HDM201 could elevate the P53 protein levels even in cells with RPL11 knockdown. These findings suggest that actinomycin D induces cell death in *TP53* wild-type BCP-ALL with P53 activation via RPL11. The inhibition of ribosomal RNA is known to be a cause of nucleolar stress response. Quantitative real-time PCR showed that actinomycin D suppressed *45S pre-rRNA* mRNA, a precursor of ribosomal RNA, in NALM6 and RS4;11 cells (Figure 1e). Immunofluorescence showed that nucleophosmin was disassembled into the nucleoplasm in NALM6 and RS4;11 cells treated with actinomycin D (Figure 1f). Using immunoprecipitation, RPL11 bound to MDM2 increased in the NALM6 cells treated with actinomycin D as compared to that in cells treated with DMSO (Figure 1g). The findings for immunofluorescence and immunoprecipitation indicate the possibility that actinomycin D induces the re-localization of nucleolus caused by the inhibition of rRNA synthesis, and then RPL11 spread to the nucleoplasm combines with MDM2 protein.

These results were consistent with the observation that actinomycin D induces the nucleolar stress response in BCP-ALL cell lines. Notably, actinomycin D induced the nucleolar stress response in BCP-ALL cells at concentrations < 2 nM. In other cancers, a low concentration of actinomycin D (<5 nM) has been reported to selectively inhibit RNA polymerase I-dependent transcription and ribosomal biogenesis, resulting in the induction of the nucleolar stress response without DNA damage [15,16].

### 3.2. Sensitivity to 6-Mercaptopurine, Methotrexate, Daunorubicin, and Cytarabine

We found that actinomycin D can activate the nucleolar stress response in BCP-ALL cell lines. We tested the sensitivity of these cell lines to the drugs used for pediatric BCP-ALL treatment. NALM6 cells with RPL11 knockdown showed resistance to 6-mercaptopurine, methotrexate, daunorubicin, cytarabine, and L-asparaginase compared to NALM6 cells transfected with scramble siRNA. RS4;11 with RPL11 knockdown showed resistance to 6-mercaptopurine, methotrexate, daunorubicin, cytarabine, and vincristine as compared to RS4;11 cells transfected with scramble siRNA. Sensitivity to 6-mercaptopurine, methotrexate, daunorubicin, and cytarabine was commonly affected by RPL11 knockdown in NALM6 and RS4;11 (Figure 2a,b).

These results suggest that RPL11 inhibition may be related to resistance to 6-mercaptopurine, methotrexate, daunorubicin, and cytarabine. We chose these four drugs for further investigation and eliminated vincristine and L-asparaginase because of the variation in sensitivity observed among cell lines.

We assessed caspase to evaluate whether these drugs can induce apoptosis in cells. Apoptosis of BCP-ALL cell lines with RPL11 knockdown was reduced in 6-mercaptopurine, methotrexate, daunorubicin, and cytarabine (Figure 2c). These results indicate that these four drugs induced apoptosis via RPL11.

### 3.3. P53 Pathway Activation via RPL11

To determine whether these four agents can induce cell death via RPL11 through the P53 pathway, the expression of P53 protein and P53-related mRNA was examined by Western blotting and reverse transcription-PCR. These four drugs increased P53 protein expression in BCP-ALL cell line cells transfected with scramble siRNA; however, RPL11 knockdown inhibited the expression of P53 protein (Figure 3a and Appendix A). The mRNA expression of *P21*, *MDM2*, and *Fas*, which are downstream genes targeted by P53, increased in cells transfected with scramble siRNA, but they did not increase in the cells with RPL11 knockdown (Figure 3b).

These results suggest that 6-mercaptopurine, methotrexate, daunorubicin, and cytarabine induce inhibition of cell proliferation and cell death via P53 pathway activation induced by the nucleolar stress response.

### 3.4. Suppression of 45S pre-rRNA mRNA Expression Followed by Disassembly of the Nucleolar and Protein Binding

Similar to actinomycin D, 6-mercaptopurine, methotrexate, daunorubicin, and cytarabine suppressed *45S pre-rRNA* mRNA after a few hours of treatment in NALM6 cells and RS4;11 (Figure 4a) following disassembly of the nucleolus (Figure 4b). These four drugs increased the binding of RPL11 to MDM2 in NALM6 cells (Figure 4c). These findings indicate the possibility that 6-mercaptopurine, methotrexate, daunorubicin, and cytarabine induce the disassembly of nucleolus caused by the inhibition of rRNA synthesis; then, RPL11 is translocated to the nucleoplasm and binds to MDM2.

### 3.5. RPL11 mRNA Expression in Leukemia Cells of Patients

We examined *RPL11* mRNA expression in leukemia cells collected from the bone marrow of pediatric patients with BCP-ALL with relapse. Although leukemia cells were not isolated and purified from the bone marrow samples, each bone marrow sample was treated as including leukemia cells because the majority of nucleated cells in the bone marrow samples were leukemia cells. The patient characteristics are shown in Appendix A. *RPL11* mRNA expression in leukemia cells at relapse decreased in seven out of nine patients compared to that at diagnosis. The level of *RPL11* mRNA expression at the second relapse was also lower than that at the first relapse (Figure 5a).

### 3.6. Resistance to 6-Mercaptopurine, Methotrexate, Daunorubicin, and Cytarabine

Using the patient’s leukemia cells (Pt 9), we investigated whether nucleolar stress was involved in the acquisition of resistance. The patient’s *RPL11* mRNA and RPL11 protein levels clearly decreased at relapse as compared to those at diagnosis (Figure 5a,b). We compared the sensitivity to 6-mercaptopurine, methotrexate, daunorubicin, and cytarabine in the patient’s leukemia cells at diagnosis and relapse and found that the patient’s leukemia cells at relapse were more resistant to the four agents compared to the cells at diagnosis (Figure 5c). In contrast, HDM201 had a similar effect on cells at relapse and diagnosis. After treating the patient’s leukemia cells with 6-mercaptopurine, methotrexate, daunorubicin, and cytarabine, P53 protein levels increased in the cells at diagnosis; however, induction of P53 was greatly suppressed in the cells at relapse (Figure 5d). Conversely, HDM201 could increase to the same level as the P53 protein in the patient’s leukemia cells either at diagnosis or at relapse.

These results suggest that insufficient nucleolar stress response in patients with relapsed BCP-ALL may be related to multidrug resistance.

## 4. Discussion

Our findings suggest that the observed decrease in RPL11 expression in pediatric BCP-ALL cells is related to resistance to 6-mercaptopurine, methotrexate, daunorubicin, and cytarabine due to RPL11 expression, which is the main regulator of the suppression of the nucleolar stress response and decreased at relapse at high frequencies. Each drug is essential in the treatment of pediatric BCP-ALL, and it can be inferred that drug resistance is associated with relapse. These drugs have not been reported to induce a nucleolar stress response in leukemia or in other cancers. The present study revealed that these four drugs could activate the nucleolar stress response as a novel anti-tumor mechanism of action.

Interestingly, 6-mercaptopurine is a precursor of 6-thioguanine nucleotide and is incorporated into DNA to exert cytotoxicity; alternatively, thioinosine monophosphate impairs purine nucleotide synthesis [17]. Methotrexate inhibits thymidylate and purine synthesis by suppressing dihydrofolate reductase activity [18]. The commonly known mechanisms of action for daunorubicin are DNA damage due to inhibition of topoisomerase II, inhibition of DNA synthesis, or production of reactive oxygen species [19,20,21,22]. Cytarabine inhibits DNA synthesis by its incorporation into DNA, competing with its metabolites, and through deoxycytidine triphosphate. RNA inhibition is not a well-known anti-tumor mechanism for any of the drugs evaluated in the present study; however, approximately 50 years ago, some reports described inhibition of rRNA or RNA polymerase via these drugs. For example, 6-mercaptopurine reportedly inhibited RNA synthesis in lymphoid leukemia cells and suppressed RNA polymerase I activity [23,24], while methotrexate suppressed rRNA transcription [25]. Daunorubicin was found to inhibit RNA synthesis in the nucleolar and *45S rRNA* synthesis [26,27]. Cytarabine inhibits RNA synthesis and suppresses RNA polymerase activity [28,29,30]. These reports are consistent with data from the present study indicating that each drug suppressed *45S pre-rRNA*.

Some gene mutations related to drug resistance have been detected in the relapse of pediatric ALL, including an increased function for NT5C2, loss of function for PRPS1, and loss of function for MSH6 related to thiopurine resistance. *NR3C1* and *CREBBP* mutations are associated with glucocorticoid resistance. Decreased folic acid polyglutamate activity is related to methotrexate resistance. These mutations were found in 55% of relapsed patients, with mutations related to glucocorticoid resistance being the most common category of mutations, followed by mutations related to thiopurine resistance [31]. Cases of anthracycline resistance due to MDM2 overexpression and cytarabine resistance due to increased Akt phosphorylation have also been reported [32,33]. Most of the known mechanisms are related to mutations relevant to drug metabolism pathways; however, ribosomal protein-related mechanisms have not been reported. The present study suggests that disabling the nucleolar stress response results in resistance to various types of drugs, including anthracyclines, purine analogs, and folate antagonists, and this finding has a great impact on the treatment strategy for relapsed pediatric BCP-ALL.

As pediatric ALL treatment involves multidrug combination chemotherapy, it is important to elucidate the mechanisms of multidrug resistance and relapse. Several multidrug resistance factors, such as P-glycoprotein, multidrug resistance protein 1, and breast cancer resistance protein, have been involved in multidrug resistance in leukemia [34]. All these factors have been established as prognostic multidrug resistance factors in adult myeloid leukemia; however, many reports have suggested that they are not associated with the prognosis of pediatric ALL [35,36,37]. The dysfunction of the nucleolar stress response is an entirely new mechanism of multidrug resistance within pediatric BCP-ALL.

Maintenance therapy for pediatric ALL is based on receiving 6-mercaptopurine and methotrexate for approximately 1–2 years. In a clinical trial that shortened the duration of maintenance therapy or reduced the dose of methotrexate, the relapse rate statistically increased [38]. Additionally, patients with higher activity of the enzyme that metabolizes 6-mercaptopurine had more relapses [17]. These clinical findings suggest that resistance to 6-mercaptopurine and methotrexate is one of the most important risk factors for relapse. In the present study, decreased expression of RPL11 drastically reduced sensitivity to 6-mercaptopurine and methotrexate in BCP-ALL cell lines and leukemia cells. Our findings suggest that decreased RPL11 expression suppresses the effect of the 6-mercaptopurine/methotrexate combination maintenance therapy and relates to relapse.

As almost all pediatric BCP-ALL cases have wild-type *TP53*, the nucleolar stress response appears to be an effective therapeutic target. RPL11 expression can be used as a biomarker to evaluate the effects of nucleolar stress response therapy. For patients with reduced RPL11 expression at the time of relapse, a combination of drugs that are effectively independent of the nucleolar stress response may be a therapeutic strategy to overcome multidrug resistance. For patients without reduced RPL11 expression, drugs that specifically induce the nucleolar stress response may be effective against pediatric ALL. As nucleolar stress response-specific therapy is supposed to activate P53 without DNA damage, nucleolar stress response-specific therapy has a significant advantage for children to avoid developing secondary cancers. To our knowledge, there has been no clinical study of pediatric leukemia investigating the efficacy of the nucleolar stress response thus far; however, Gionfriddo et al. [39] recently reported that the nucleolar stress response induced by actinomycin D was effective among adult patients with relapsed/refractory acute myeloid leukemia. This report suggests that the nucleolar stress response may be a new therapeutic target for pediatric leukemia for patients with BCP-ALL.

*TP53* mutations occur in a minor subtype of pediatric BCP-ALL; however, overcoming this subset of mutations is essential to eradicating pediatric BCP-ALL. A previous report describes that the mechanism of the nucleolar stress response acts independently of P53 [40]. It is considered that P27 and RPL3 released from the nucleolus directly activate P21 or suppress CDK4/6, followed by cell cycle arrest or apoptosis, without RPL11 and P53 [41,42]. Although we did not assess the P53-independent nucleolar stress mechanism in the present study, we recommend that future studies should examine whether this mechanism affects BCP-ALL cells.

The current study has some limitations. First, only two cell lines were used. As the *TP53* wild-type ALL cell line was limited, we could not conduct this study using several cell lines. Second, the present study had no in vivo assessment using animals. Instead of in vivo assessment, we used patient samples to compare drug sensitivity with respect to diagnosis and relapse via a protein assay. In addition, few patients were enrolled in the present study. To determine whether the nucleolar stress response is related to relapse, a larger number of pediatric patients with BCP-ALL with relapse need to be assessed.

## 5. Conclusions

We identified a novel anti-tumor mechanism (the nucleolar stress response) with respect to 6-mercaptopurine, methotrexate, daunorubicin, and cytarabine. As these drugs are used for pediatric patients with BCP-ALL for their intended effects on the nucleolar stress response, dysfunction of the nucleolar stress response may be related to multidrug resistance. By developing a test system to measure the RPL11 expression, multidrug resistance at relapse can be predicted and used for therapeutic stratification in relapsed pediatric BCP-ALL cases. Furthermore, if drugs that upregulate the RPL11 expression and enhance the function of the nucleolar stress response are identified, therapeutic strategies for overcoming multidrug resistance in pediatric BCP-ALL can be developed. Thus, nucleolar stress response-targeting therapy is expected to be an effective and ideal therapy for pediatric patients with BCP-ALL.

## Figures and Tables

**Figure 1 cancers-14-05127-f001:**
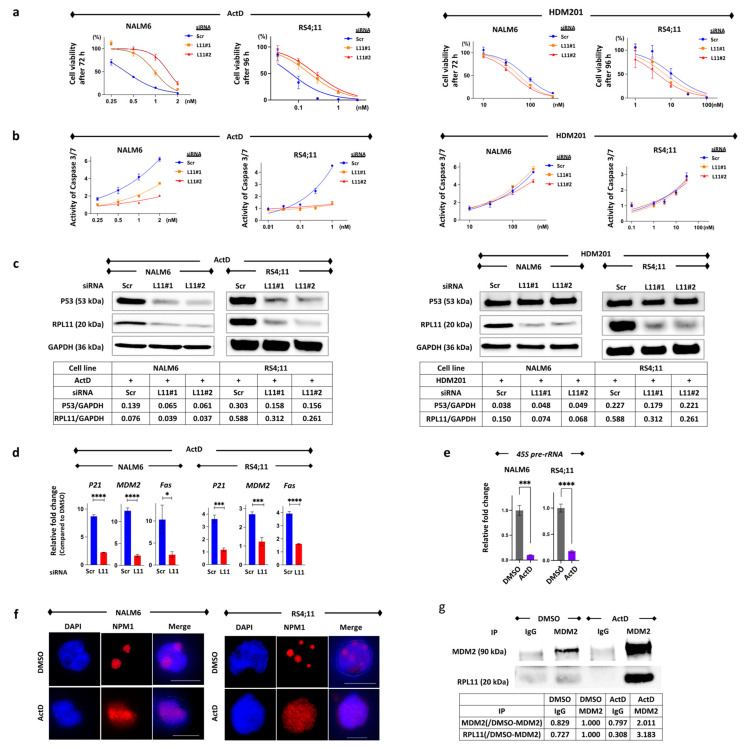
Actinomycin D induces cell death in B-cell precursor acute lymphoblastic leukemia cell lines via nucleolar stress response. (**a**) Drug sensitivity to actinomycin D (ActD) and HDM201 was assessed by examining cell viability after treatment with drugs for 72 h in NALM6 cells (left) and 96 h in RS4;11 (right) with transfected scramble (Scr) or siRNA against RPL11 (L11#1 and L11#2). Cell viability was measured using an ATP-dependent bioluminescence assay. Cell viability was calculated based on the transfected cells treated with the same concentration of DMSO as ActD or HDM201. Reactions were carried out in quadruplicate. (**b**) Caspase 3/7 activity was measured via the Caspase Glo^®^ 3/7 assay after treatment with drugs for 24 h in NALM6 cells (left) and 48 h in RS4;11 (right) with transfected scramble or siRNA against RPL11 (L11#1 and L11#2). Reactions were carried out in triplicate. (**c**) Western blot analysis of P53 and RPL11 at 24 h after 1 nM ActD or 100 nM HDM201 treatment in NALM6 cells as well as after 48 h of 0.1 nM ActD or 10 nM HDM201 treatment for RS4;11 transfected with scrambled siRNA (Scr) or RPL11 siRNA#1 or RPL11 siRNA#2 (L11#1 and L11#2). The GAPDH levels were used as loading control. (**d**) Relative fold changes of P53-targeting gene mRNA expression after treatment with 1 nM ActD for 24 h in NALM6 cells and 0.25 nM ActD for 48 h in RS4;11. The blue bar indicates cells transfected with scramble siRNA (Scr); the red bar indicates cells transfected by *RPL11 siRNA* (L11). Error bars represent standard deviation (*n* = 3, * *p*-value < 0.05; *** *p*-value < 0.001; **** *p*-value < 0.0001). (**e**) The *45S pre-rRNA* mRNA expression of NALM6 cells and RS4;11 after treatment with DMSO or 1 nM ActD for 24 h. Error bars represent standard deviation (*n* = 3, *** *p*-value < 0.001; **** *p*-value < 0.0001). (**f**) Typical immunofluorescence images were obtained after staining NALM6 and RS4;11 cells with antibodies specific to nucleophosmin (NPM1). Chromatin was stained using DAPI. NALM6 cells were treated with DMSO or 1 nM ActD, and RS4;11 cells were treated with DMSO or 0.25 nM ActD. (**g**) Immunoblotting by anti-RPL11 after immunoprecipitation with control IgG or anti-MDM2 antibody for NALM6 cell lysate treated with DMSO or 1 nM ActD for 24 h. The cell lysates were immunoprecipitated with control IgG or anti-MDM2 antibody followed by immunoblotting to detect RPL11. ActD, actinomycin D; DMSO, dimethyl sulfoxide; IgG, immunoglobulin G; NPM1, nucleophosmin.

**Figure 2 cancers-14-05127-f002:**
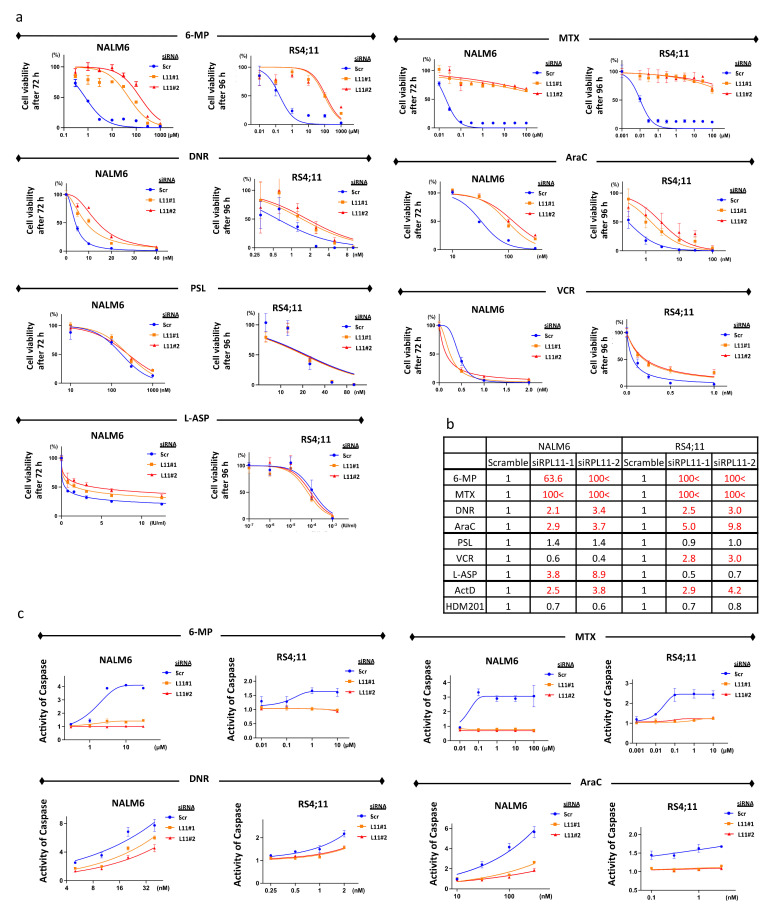
RPL11 inhibition induces resistance to 6-mercaptopurine, methotrexate, daunorubicin, and cytarabine. (**a**) Drug sensitivity was assessed using cell viability after treatment with drugs for 72 h in NALM6 with transfected siRNA and for 96 h in RS4;11 with transfected siRNA. Cell viability was measured using an ATP-dependent bioluminescence assay. Reactions were carried out in triplicate. Cell viability was calculated based on the transfected cells treated with the same concentration of DMSO or water as each drug. Error bars represent standard deviation. (**b**) Fold change of IC_50_ normalized by scramble. (**c**) Caspase activity was measured via the Caspase Glo^®^ 3/7 assay after treating with drugs for 24–48 h. Error bars represent standard deviation. 6-MP, 6-mercaptopurine; AraC, cytarabine; DNR, daunorubicin; L-ASP, L-asparaginase; L11#1, RPL11 siRNA#1; L11#2, RPL11 siRNA#2; MTX, methotrexate; PSL, prednisolone; Scr, scrambled siRNA; VCR, vincristine.

**Figure 3 cancers-14-05127-f003:**
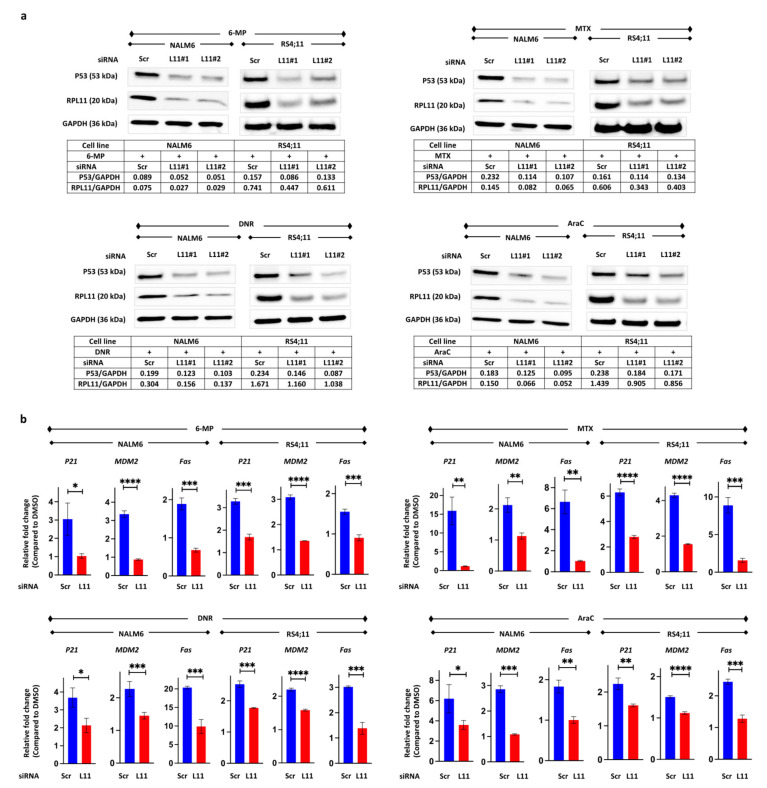
6-mercaptopurine, methotrexate, daunorubicin, and cytarabine induce the P53 pathway activation. (**a**) Western blot analysis of P53 and RPL11 after 24 h of drug treatment in NALM6 cells and 48 h of drug treatment in RS4;11 cells transfected with scrambled siRNA or RPL11 siRNA#1 or #2. GAPDH levels were used as a loading control. Each cell was treated with 100 μM 6-MP, 100 nM MTX, 10 nM DNR, and 100 nM AraC for NALM6 cells as well as with 10 μM 6-MP, 100 nM MTX, 3 nM DNR, and 3 nM AraC for RS4;11 cells. (**b**) P53 target gene mRNA expression was measured via reverse transcription-polymerase chain reaction. Each cell was treated with the same concentration of each drug as Western blot for 24 h in NALM6 cells and 48 h in RS4;11 cells. Reactions were carried out in triplicate. The blue bar indicates cells with transfected scramble siRNA; the red bar indicates the cells transfected with RPL11 siRNA. Error bars represent standard deviation (* *p*-value < 0.05; ** *p*-value < 0.005; *** *p*-value < 0.001; **** *p*-value < 0.0001). 6-MP, 6-mercaptopurine; AraC, cytarabine; DNR, daunorubicin; DMSO, dimethyl sulfoxide; L11#1, RPL11 siRNA#1; L11#2, RPL11 siRNA#2; MTX, methotrexate; Scr, scrambled siRNA.

**Figure 4 cancers-14-05127-f004:**
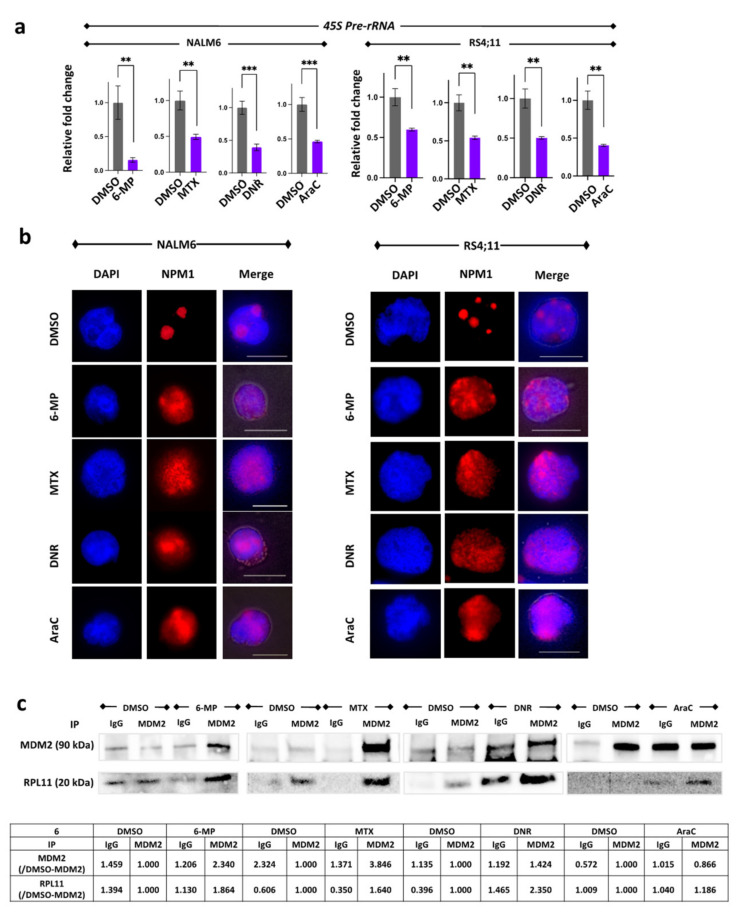
Suppression of *45S pre-rRNA* mRNA expression followed by disassembly of the nucleolus and protein binding. (**a**) This image depicts *45S pre-rRNA* mRNA expression after treatment with each drug. Each cell was treated with 100 μM 6-MP, 100 nM MTX, 10 nM DNR, and 100 nM AraC for NALM6 cells and with 10 μM 6-MP, 100 nM MTX, 3 nM DNR, and 3 nM AraC for RS4;11 cells. Error bars represent standard deviation (** *p*-value < 0.005; *** *p*-value < 0.001). (**b**) Typical immunofluorescence images were obtained after staining NALM6 cells and RS4;11 cells with antibodies specific for nucleophosmin (NPM1). Chromatin was counterstained using DAPI. Cells were treated with 100 μM 6-MP, 100 nM MTX, 10 nM DNR, and 100 nM AraC for NALM6 cells and with 10 μM 6-MP, 100 nM MTX, 3 nM DNR, and 3 nM AraC for RS4;11 cells. (**c**) Immunoblot of NALM6 cells treated with each drug for 24 h. Cell lysates were immunoprecipitated with control IgG or antibody to MDM2 followed by immunoblotting to detect RPL11. Each cell was treated with 3 μM 6-MP, 30 nM MTX 10 nM DNR, and 100 nM AraC. 6-MP, 6-mercaptopurine; AraC, cytarabine; DMSO, dimethyl sulfoxide; DNR, daunorubicin; IgG, immunoglobulin G; MTX, methotrexate; NPM1, nucleophosmin.

**Figure 5 cancers-14-05127-f005:**
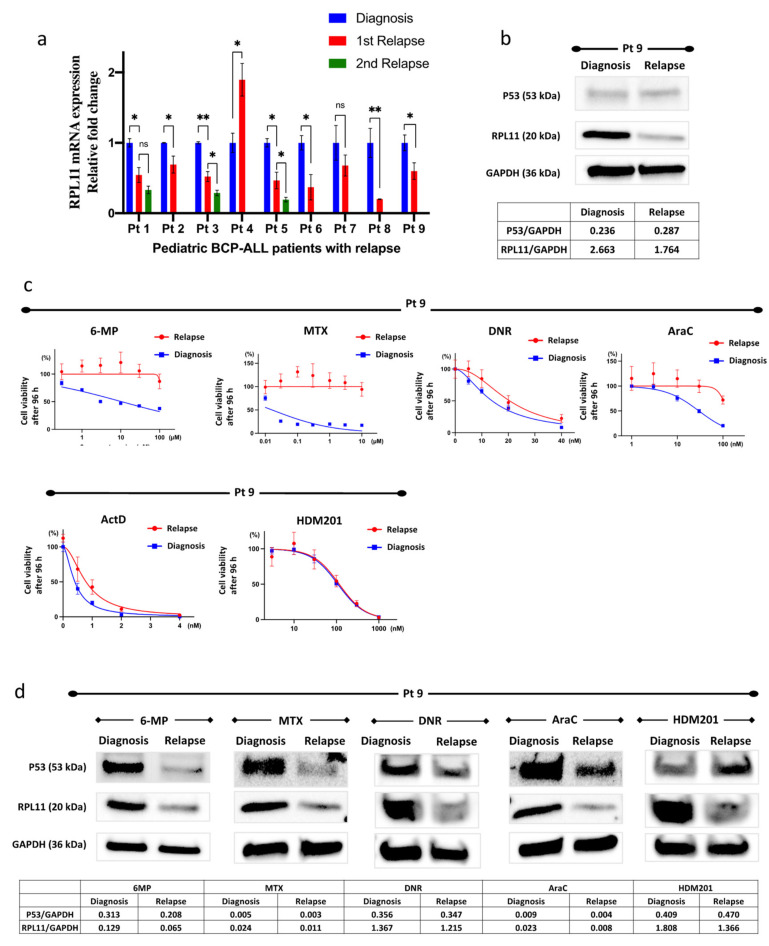
Analysis of patients’ leukemia cells. (**a**) *RPL11* mRNA expression of leukemia cells from patients with pediatric B-cell precursor acute lymphoblastic acute leukemia (BCP-ALL). The graph compares changes at relapse to those at diagnosis for each patient. Error bars represent standard deviation (* *p*-value < 0.05; ** *p*-value < 0.005). (**b**) Immunoblot of leukemia cells at diagnosis and the relapse of Patient 9 without drug stimulation. (**c**) Drug sensitivity was measured using cell viability after treating with drugs for 96 h in leukemia cells, at diagnosis, and at relapse in Patient 9, using an ATP-dependent bioluminescence assay. Cell viability was calculated based on the cells treated with the same concentration of DMSO as each drug. Reactions were carried out in triplicate. Error bars represent standard deviation. (**d**) Western blot analysis of P53 and RPL11 following drug treatment in leukemia cells at diagnosis and relapse (Patient 9). The GAPDH levels were used as a loading control. Each sample was treated with 10 μM 6-MP for 48 h, 100 nM MTX for 48 h, 20 nM DNR for 24 h, 100 nM AraC for 24 h, and 100 nM HDM201 for 24 h. 6-MP, 6-mercaptopurine; ActD, actinomycin D; AraC, cytarabine; DNR, daunorubicin; MTX, methotrexate; Pt, patient; ns, not significant.

## Data Availability

No new data were created or analyzed in this study. Data sharing is not applicable to this article.

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
