# Peer review of "Association between Dysfunction of the Nucleolar Stress Response and Multidrug Resistance in Pediatric Acute Lymphoblastic Leukemia"

_cancers, 2022, doi:10.3390/cancers14205127_

Round 1
Reviewer 1 Report
In the manuscript „Association between dysfunction of the nucleolar stress response and multidrug resistance in pediatric acute lymphoblastic leukemia“ authors report a novel anti-tumor mechanism, the nuclear stress response, which could be responsible for multidrug resitance of pediatric BCP-ALL. They evaluated the response of BCP-ALL cell lines and primary samples to several drugs tested regarding cell viability, apoptosis and p53 signaling pathways. Their focus of interest was ribosomal protein 11 which possibly translocates from the nucleolus to nucleoplasm upon stimulation with chemotherapeutic drugs and inhibition of rRNA synthesis. When downmodulated, the authors observed a decrease in apoptosis in response to therapy and increase in relapse in primary patients’ samples.
Broad comments: The manuscript is interesting to read and reports on the important matter, however there are some suggestions which should increase the value of the manuscript.
Specific comments:
1. Line 136 – why different time points between NALM6 and RS4;11 cells? Please indicate time points in the Figures
2. Figure 1a – in respect to which parameter is the cell viability mentioned? It would be more informative to also have the viability of untreated cells (eg. cells in the culture that are control cells and nor treated with ActD nor transfected with scrambled RNA. Same applies to all other figures.
3. Fig 1c – The authors mention the increased level of P53 when cells are treated with ActD but there are no untreated control cells to compare the expression of TP53, RLP11 etc. Please include control untreated cells - the same applies to WB in all other figures (eg. Fig3, Fig5d).
4. Immunoblots are poor quality, especially Fig1c RLP11 for DMSO, AraC, 6-MP and MTX, and Fig5d TP53 for MTX and AraC
5. Indicate molecular weight on WB
6. Line 368-374 – insert/check references.
7. I would advise to do the flowcytometric cell cycle analysis of at least these two cell lines, treated with drugs, and to check if there is any difference in cell cycle when you downmodulate RLP11.
Reviewer 2 Report
The manuscript deals with drug resistance in leukemia cells mediated by mechanisms related to nucleolar/ribosomal stress. Authors present new original data convincing that in childhood B-cell precursor acute lymphoblastic leukemia relocation of ribosomal protein L11 (RPL11) from nucleolus might mediate sensitivity to cytostatic drugs like 6-mercaptopurine, methotrexate, daunorubicin, and cytarabine. The manuscript is richly illustrated and includes extensive data obtained on cell lines and bone marrow materials from patients. The article is of absolute interest to readers of the Cancers journal. The validity of the results is beyond doubt. However addressing few minor points might improve the presentation.
1 at line 311 authors state thet "RPL11 mRNA expression in leukemia cells at relapse decreased in seven out of nine patients compared to that at diagnosis." Only patients with relapse are included in the study. It would be interesting to compare RPL11 mRNA expression levels at diagnosis between patients with relapse and patients successfully treated (w/o relapse) with 6-mercaptopurine, methotrexate, daunorubicin, and cytarabine.
2 Authors do not discuss how they are confident, that RPL11 mRNA expression was measured in leukemic cells and not total bone marrow mononuclear cells. Do they attempted any cell sorting? What was the blast counts in bone marrow samples? Do they have any reference data concerning RPL11 mRNA expression in the samples from bone marrow donors?
3 At line 436 authors conclude that "By developing a test system to measure the RPL11 expression, multidrug resistance can be predicted and used for therapeutic stratification in pediatric BCP-ALL." This conclusion is not supported by the data presented. Authors have shown decreased level of RPL11 expression measured at relapse compared to that measured at diagnosis. To predict resistance and stratify patients one need the data available before treatment.
Round 2
Reviewer 1 Report
I have no further suggestions.